# New Therapeutic Perspectives in Prostate Cancer: Patient-Derived Organoids and Patient-Derived Xenograft Models in Precision Medicine

**DOI:** 10.3390/biomedicines11102743

**Published:** 2023-10-10

**Authors:** Vittoria Rago, Anna Perri, Silvia Di Agostino

**Affiliations:** 1Department of Pharmacy, Health and Nutritional Sciences, University of Calabria, 87036 Rende, Italy; 2Department of Experimental and Clinical Medicine, Magna Graecia University, 88100 Catanzaro, Italy; anna.perri@unicz.it; 3Department of Health Sciences, Magna Græcia University of Catanzaro, 88100 Catanzaro, Italy

**Keywords:** prostate cancer, patient-derived organoid (PDO), patient-derived xenograft (PDX), precision medicine, targeted therapy

## Abstract

One of the major goals in the advancement of basic cancer research focuses on the development of new anticancer therapies. To understand the molecular mechanisms of cancer progression, acquired drug resistance, and the metastatic process, the use of preclinical in vitro models that faithfully summarize the properties of the tumor in patients is still a necessity. The tumor is represented by a diverse group of cell clones, and in recent years, to reproduce in vitro preclinical tumor models, monolayer cell cultures have been supplanted by patient-derived xenograft (PDX) models and cultured organoids derived from the patient (PDO). These models have proved indispensable for the study of the tumor microenvironment (TME) and its interaction with tumor cells. Prostate cancer (PCa) is the most common neoplasia in men in the world. It is characterized by genomic instability and resistance to conventional therapies. Despite recent advances in diagnosis and treatment, PCa remains a leading cause of cancer death. Here, we review the studies of the last 10 years as the number of papers is growing very fast in the field. We also discuss the discovered limitations and the new challenges in using the organoid culture system and in using PDXs in studying the prostate cancer phenotype, performing drug testing, and developing anticancer molecular therapies.

## 1. Introduction

Prostate cancer (PCa) is the most commonly diagnosed malignancy and the second leading cause of death in males [1,2]. Risk factors for this condition include older age, family history, obesity, hypertension, sedentary lifestyle, high testosterone levels, and ethnicity [3]. There are generally no initial or early symptoms, but late symptoms may include renal failure from ureteral obstruction, anemia, bone pain, and paralysis from spinal metastases.

The primary diagnosis is made by prostate-specific antigen (PSA) testing, although the use of this test for screening remains controversial [4]. On the other hand, ultrasound-guided transrectal prostate tissue biopsies (TRUS) are significant [5]. The latest diagnostic tests include the prostate cancer gene3 (PCA3) urine test, measurement of free and total PSA levels, prostate health index (PHI) score, 4K score/multiparametric magnetic resonance imaging (4K/MRI test), exosome test, genomic analysis, magnetic resonance imaging (MRI), the prostate imaging–reporting and data system (PIRADS), and prostate biopsy using MRI fusion techniques [6]. When cancer is localized, it is considered curable; when it spreads outside the prostate, treatment approaches include hormone treatment, chemotherapy, radiopharmaceuticals, immunotherapy, pain medications, bisphosphonates, rank ligand inhibitors, focused radiation, and other targeted therapies [7]. The effects of these therapies depend on the age of the patient, any associated health problems, the histology of the tumor, and the extent of the disease [8].

PCa treatment usually involves androgen deprivation (ADT), surgical therapy, and pharmaceutical management [1,2]. Androgen receptor signaling plays a key role in the development and proliferation of PCa; for this reason, medical castration by androgen deprivation therapy (ADT) represents a mainstay in standard-of-care PCa. However, long-term ADT, in many patients, determines forms of castration-resistant PCa (CRPC), which is manifested by an increase in the levels of prostate-specific antigen despite castration [9]. Despite different treatment options for CRPC, such as hormone therapy, immunotherapy, chemotherapy, radionuclides, and the use of targeted therapies such as PARP inhibitors, the metastatic CRPC (mCRPC) remains a lethal disease with a very low survival rate [9]. The latest PCa treatments are based on the knowledge of predictive molecular signatures to design an optimal targeted therapeutic protocol [10]. To support these experimental therapies, recent studies have indicated PCa-related genetic abnormalities associated with CRPC (i.e., AR, TP53, PTEN, and BRCA 1/2) [10].

However, due to limited access to tumor tissue for molecular profiling, progress on the use of these new therapeutic approaches is limited. Furthermore, to date, there are no diagnostic approaches that understand the heterogeneity of PCa to different diseases [10].

## 2. Patient-Derived Preclinical Models

Different preclinical PCa models have been developed to identify the molecular mechanisms underlying cancer progression and treatment resistance. Three PCa cell lines, namely LNCaP, DU-145, and PC3, are commonly used for in vitro studies [11], but although these cells allow identifying predictors of treatment response and resistance, they do not represent the diversity and the heterogeneity of human prostate cancer. Therefore, in order to identify new therapeutic targets and make progress in oncology, it is necessary to deepen the cellular and functional biology of the tumor. The need has therefore developed to study cancer no longer on 2D cell lines but in patient-derived models that can faithfully summarize and capture the complexity and heterogeneity of the tumor ecosystem. The creation of omics research groups that facilitate the study and analysis of individual patient tumors are crucial.

A patient-derived preclinical model, also known as the patient-derived xenograft (PDX) model, is a research tool used in cancer studies. It involves transplanting tumor tissue directly from a patient into an immunodeficient mouse or other animal models. This allows the tumor to grow and develop in the animal, creating a model that closely mimics the characteristics of the original patient’s tumor [12]. PDX represents a useful preclinical model for overcoming limitations associated with the use of cancer cell lines, since it retains the key molecular aberrations present in patient tumors that are involved in the tumoral growth. In addition, PDXs have the advantage of partly recreating the complexity of the tumor microenvironment (TME), providing a more reliable and clinically predictive model for drug response screening.

However, the most important limitation of PDX is the lack of influence of the immune system as the mice are immunodepleted. Indeed, although the development of severe immunodeficient mice improved the PDX take rate, these models have a limitation of application because the metastatic behavior of cancer cells in severe immunodeficient models differs from the clinical situation. However, the development of “humanized NSGs” showing functional human T cells, NK cells, and monocytes is helping to better understand the role of cancer immunology and the drug response in the context of immune sufficiency [13]. Furthermore, the application of chimeric grafts with neonatal mouse mesenchyme has significantly increased the xenograft survival rate and doubled the proliferation index of xenografted cancer cells [14]. Most importantly, neuroendocrine prostate cancer (NEPC) PDX models have facilitated the interpretation of NEPC biology and drug sensitivities, allowing the matching of the molecular characteristics of subgroups of NEPC with specific targeted therapeutics. In addition, these preclinical models exhibit a significant role in investigating neuroendocrine transdifferentiation, which has been recognized as a therapeutic resistance mechanism [15]. However, despite this encouraging progress, it is crucial to have an extensive collection of NEPC PDX models covering a range of heterogeneities.

Patient-derived organoids (PDO) are three-dimensional cultures of cells derived from patient tumor samples. They are generated by isolating and culturing tumor cells or tissue fragments in a specialized culture system that supports the growth and self-organization of cells into structures that resemble miniature organs or tissues [16]. In particular, the organoids have provided exciting new opportunities for translational research in oncology, enabling the development of patient-representative biobanks that enable testing for personalized therapies and predict responses prior to their use on the patient.

Organoid cultures from CRPC biopsy samples exhibit highly variable growth rates, presumably depending on the heterogeneity of phenotypes observed in patients; in addition, the long time required for CRPC organoids to expand limited their use to test therapeutic responsiveness. Alternatively, the availability of large CRPC organoid cohorts could be useful in identifying classes of responders and non-responders, as well as potential predictive biomarkers. Organoid cultures can be obtained also from PDX, whose success rate is higher than from biopsy (~50%). Cultures of CRPC PDX-derived organoids are models exhibiting genetic and phenotypic heterogeneity, and therefore, results are particularly useful for mechanistic studies, biomarker identification, and evaluation of drug responses [16]. Furthermore, the co-culture of organoids with immune cells isolated from autologous tumor tissues or peripheral blood of healthy donors represents an attractive approach to exploring the interaction between the immune system and tumors and to predicting the response of patients to immunotherapy [17].

A scheme of the workflow for preparing all these patient cell models is summarized in Figure 1.

### 2.1. Prostate Cancer PDX Models

PDXs are preclinical models that involve transplanting tumor tissue from prostate cancer patients into immunodeficient mice. PCa PDX models capture the heterogeneity of the disease, reflecting the genetic and molecular diversity reported in patients [18,19]. These models are useful in cancer research to study the characteristics of prostate cancer, test potential treatments, and develop personalized therapeutic strategies [20,21].

The duration of tumor settlement in mice can take from a few weeks to a few months to observe the tumor nodule (F0: first generation). After serial transplantation (F1, F2, Fn), the duration of tumor growth becomes stable, and the days needed to obtain tumors of a discrete size depends on the aggressiveness of the PCa [22,23]. Usually, F2/F3 PDX tumors are suitable for cancer biology studies, such as drug sensitivity screening and biomarker identification. The success rate of PDX stabilization varies according to the origin of the tumor and the characteristics of the disease; in PCa, the rate is 65–70% [22,23]. Although the advantage of PDX is to maintain a tumor microenvironment, the loss of human tumor stroma is already observed in the second generation with complete replacement by mouse stroma [24]. The mouse xenograft models that have been used for years are based on immunocompromised mice to avoid rejection of the tissue or injected human tumor cells. Therefore, these mice were found to be inadequate to study the TME in PDXs. To create models versatile in stabilizing a human PDX tumor model, genetically humanized mice and immunologically humanized mice have been developed [13,25].

At the preclinical level, PDXs can be used to identify and validate biomarkers associated with prostate cancer. By comparing the genetic and molecular characteristics of PDX tumors with patient data, several studies reported prediction of treatment response, disease progression, or prognosis [26,27]. Currently, PDX models are the most clinically interesting and adherent in vivo cancer models of drug response obtained from cancer patients [18,28,29]. Thus, the NCI (National Cancer Institute of US) had the initiative to replace a panel of 60 human cell lines (NCI-60) with PDXs for drug testing and the screening of molecules for therapeutic purposes [30].

However, current PCa PDXs, and more generally PDX models of all tumor types, cannot be considered perfectly fitting “avatars” of human cancer. Although PDX in humanized mice for several types of tumors including PCa have been developed, human stromal components of implanted tumor tissue are rapidly lost and replaced by mouse TME during engraftment [31,32,33]. Recently, considering 1110 PDX samples across 24 cancer types, it was documented that PDXs showed rapid accumulation of chromosomal copy number aberrations (CNAs) during the passages, often because of selection of different preexisting cellular clones. CNA acquisition in PDXs was correlated with the genetic heterogeneity observed in primary implanted tumors. Interestingly, some alterations of the copies acquired during PDX passaging differed from those acquired in patients in the course of the disease [33].

Of notable foresight has been a project initiated in 1996 by a group of researchers at the MD Anderson Cancer Center that involved a group of clinically annotated patient-derived xenografts linked to specific phenotypes reflecting all aspects of prostate cancer [34]. The study morphologically characterized 80 PDXs derived from 47 human prostate cancer donors; 47 PDXs derived from 22 donors were working models and were found to be suitable for the growth as cell lines (MDA-PCa 2a and 2b cells) or PDXs [34]. In detail, these PDXs maintained the phenotypic and molecular features of the original tumor, reacting in an expected way to mouse castration and to targeted molecular treatments. Interestingly, supporting the utility of the PDX model in studying tumor heterogeneity, pairs of PDXs derived from different areas of the same tumor showed significant differences in oncogenic pathways, as verified by genomic and RNA sequencing analyses [34] (Table 1).

Very recently, five new PCa PDX models were characterized, including hormone-naïve, androgen-sensitive, and CRPC primary tumors and prostate cancer with neuroendocrine differentiation (NEPC), which is a rare subtype of PCa, and it can either arise de novo or arise as resistance to therapies [35,36]. This biobank was characterized by a comprehensive genomic analysis leading to the identification of driver alterations in androgen signaling, DNA repair, and PI3K pathways, and it was evaluated for androgen deprivation, PARP inhibitors, and chemotherapy response [35]. Overall, these PDXs recapitulated the molecular subtypes of prostate cancer.

**Table 1 biomedicines-11-02743-t001:** Benefits and drawbacks of PDX and PDO human-patient-derived models.

Models	Benefits	Drawbacks
Mouse PDX-patient-derived Xenografts [12,18,19,25,26,27,28,29,34]	Possibility to develop the tumor in a physiological TME	Need for an animal house and high costs for maintaining the mice
	Possibility to study tumor cell heterogeneity in vivo	Long time for engraftment experiments
	Crosstalk between factors of the murine immune system and the tumor	High failure rate in engraftment
	Possibility to study the response to therapies with in vivo parameters	
PDO-patient-derived Organoids [20,36,37,38,39]	Limited costs for the formation and maintenance of organoids	Absence of a physiological TME
	Formation of the organoids in a few days and possibility of amplification in more avatars in the first passages	After a few passages the organoids change the molecular characteristics of the tumor of origin
	Organoid ability to grow on scaffolds and mimic signaling as in physiological TME	
	Ability to reproduce the structure of the primary tumor tissue	

All these described results provided a wealth of data and insight into the biological mechanisms that could underpin tumor cellular heterogeneity and function as an important resource for the development of personalized therapy targeting specific molecular markers of prostate cancer.

### 2.2. Prostate Cancer PDO Models

PCa PDOs are three-dimensional cell cultures that are derived from isolated pluripotent stem cells or organ progenitor cells of patient samples, such as tumor tissues or matched healthy tissues [37,38]. The first 3D cultures of patient-derived prostate epithelial cells in the equivalent of artificial basement membrane (BME) matrigel was obtained in the presence of serum, dihydrotestosterone, and stomal cells [39]. These culture conditions, dependent on the number of stem cells in the starting tissue sample, produced morphological differentiation resulting in spheroid-like acini. The first long-term culture of prostate cancer from biopsy specimens and circulating tumor cells occurred in 2014 when the authors reported the first fully molecularly characterized organoids, which recapitulated the molecular diversity of prostate cancer subtypes, including TMPRSS2-ERG fusion, SPOP mutation, SPINK1 overexpression, and CHD1 loss [37].

PCa PDOs were shown to have some disadvantages, such as low efficiency of establishing organoids (15–20% successful in establishing), lack of a physiological TME and immune system, and contamination of normal cells; however, PCa PDOs provide a valuable tool for studying the biology of prostate cancer and testing potential therapeutic strategies [20,37]. PCa PDOs can be used to investigate the genetic and molecular characteristics of individual patient tumors, including their response to different drugs or treatments, having the potential to capture the genomic and clinical heterogeneity observed in prostate cancer and showing a high correlation with the drug response of primary tumors in patients [20] (Table 1).

Prostate cancer is a highly heterogeneous disease, both at the genetic level and in terms of clinical behavior and response to treatment. This heterogeneity poses challenges in understanding the disease and developing effective treatments. Cultures of organoids from CRPC biopsy specimens demonstrate highly variable rates of formation and proliferation, likely demonstrating the heterogeneity of phenotypes observed in patients. Compared to the efficient establishment and rapid growth of organoids from other cancer types, CRPC organoids need a few weeks or even months to grow, to be characterized and used for functional testing [20,40]. This aspect makes the personalized medicine approach for individual PCa patients very complicated, limiting studies for therapeutic responses. However, once the organoid has been created, studies of associated response mechanisms and potential predictive biomarkers have increased in recent years.

Among the disadvantages of culturing organoids there is a lack of cells that populate them, such as fibroblasts, endothelial cells, and immune cells, among others, thus preventing the system from fully recapitulating the complex TME interactions. However, the 3D structures are composed of fully differentiated basal and luminal cells, rare neuroendocrine cells, and cells with stem cell characteristics [41,42]. The presence of TME is also important to determine the success of a treatment. About 90% of preclinically approved drugs do not have the desired effect in clinical trials, partially because of the use of too-simplified in vitro models and the lack of mimicking the TME in drug efficacy screening [42]. Thus, efforts focused on adding the TME components to organoids to recapitulate the microenvironment of parental tumors are ongoing.

## 3. Current Application of PDO and PDX in PCa Cancer Research

PDO and PDX models are two powerful tools used in cancer research, particularly in the field of personalized medicine and drug development. When combined with high-throughput sequencing technologies focusing on non-coding RNA, biomarkers, and proteomics, they offer a comprehensive approach to understanding cancer biology and identifying potential therapeutic targets.

PCa research has been slowed by the usual 2D in vitro models, which cannot fully reflect the biological and clinical characteristics of PCa. The PDO model in three-dimensional culture preserves the heterogeneity of primary tumor tissues and allows for high-throughput screening and genome editing. Therefore, the establishment of a PCa organoid model that recapitulates the different heterogeneity in patients is of great importance to study the PCa from a basic research perspective. The success of basic research using PCa organoids is the fine-tuning of culture techniques and conditions and the creation of a TME to provide an in vitro environment for cells or tissues to develop into “PCa avatars” [38,42,43,44] (Table 1).

The cell from which PCa originates is still a subject of study and discussion. It has been previously reported that PCa mainly has two cell types, basal and luminal cells, while in both in vitro mouse and human models, only basal cells appear to reconstitute a complete prostate gland [45,46]. However, with the development of selective purification techniques using cellular markers, both basal and luminal cells are able to produce an organoid in both mice and humans, maintaining responsiveness to androgens [47], demonstrating that both types of cells have a stem cell potential that can cause prostate cancer.

The organoids can be cultured in nonadherent conditions or embedded in prefabricated ECMs prepared from diverse materials such as natural or synthetic scaffolds [42,48]. Other strategies involve the reconstitution of TME by adding cancer-associated fibroblasts (CAFs) to PDO models [49]. CAFs can promote tumor growth through various mechanisms. They can secrete growth factors, cytokines, and extracellular matrix proteins that stimulate the growth and survival of PCa cells [50]. Additionally, they can remodel the extracellular matrix to facilitate cancer cell migration and invasion. Cancer-associated fibroblasts can also suppress the immune system’s response to the tumor, allowing cancer cells to evade immune surveillance [50,51].

The immune system is another crucial component of TME. Advances in immunotherapy based on chimeric antigen receptor (CAR) lymphocyte transfer therapies and immune checkpoint inhibition have revolutionized the treatment options of PCa [52,53,54]. Additionally, 3D hanging-drop methods are validated to culture PCa organoids as tumor epithelial monocultures and as epithelial–stromal co-cultures [43].

Recently, very interesting research has characterized prostate cancer TME by performing single-cell RNA sequencing on prostate biopsies, prostatectomy specimens, and patient-derived organoids from patients with localized prostate cancer [44]. The study characterized tumor cell heterogeneity, subpopulations of epithelial cells, stromal cells, and tumor microenvironment. The authors identified a distinctive epithelial cell population of club cells that had not previously been observed in human prostate cancer samples that may be involved in PCa carcinogenesis, and PCa-enriched epithelial cell states identified in the tumor tissues were also present on in vitro organoid cultures [44]. Consistent with the identified gene signatures and the 19 cell clusters into which the tissue samples and organoids were classified, the results suggested that club PCa cells were more androgen sensitive overall and harbor a highly androgen-responsive cellular state that can play a supportive role for the overall androgen-responsive cellular environment of prostate cancer [44].

Considering the heterogeneity of PCa, many studies are still in progress to identify the driver mutations. The strategy for exploiting PDOs should be to perform gene sequencing from organoids derived from normal and tumor tissues from the same patient that can be simultaneously established. It is very difficult to obtain organoids from primary prostate cancers [40]; however, Gao and colleagues obtained PCa PDO from advanced/metastatic stage tumors and from circulating tumor cells (CTCs) [37]. Gene sequencing revealed several gene alterations superimposable on those of the tissue analyses of the corresponding patients including SPOP mutations, PTEN loss, and TMPRSS2-ERG interstitial deletion, as well as alterations in TP53, PIK3R1, FOXA1, and several chromatin modifier mutations.

Drug resistance in cancer is often linked to mutational events in the DNA or tumor cell lineage, but the molecular mechanisms that modulate this plasticity in cell populations remain unclear. Using genetically modified mouse organoids, a recent study has provided some answers on the causes of lineage plasticity in PCa and its relation to antiandrogen resistance [55]. The authors reported that plasticity would start from epithelial cells identified by a mixed luminal–basal phenotype and that it depends on high JAK (Janus kinase) and FGFR (fibroblast growth factor receptor) activity. Significantly, PDOs with castration-resistant PCa-harboring mixed-lineage cells reproduced the addiction observed in mice by regulating the gene expression of luminal phenotype cells after treatment with JAK and FGFR inhibitors. Single-cell analysis confirms the presence of mixed-lineage cells with highly activated JAK/STAT and FGFR pathways in a subset of patients who develop metastases [55]. These data have a strong clinical impact as they provide more detail for patient stratification and therapies associated with the specific subgroups.

To date, PDX represents the most valid model to represent both the biology and heterogeneity of cancer and to simulate TME, also considering the development of humanized mice. PDXs have been extremely useful and versatile in the preclinical phase to test innovative therapies. In fact, approximately 90% of drugs that successfully pass preclinical studies by using cell lines are subsequently ineffective when they are tested in the human Phase 1–2 [56,57].

In the last years, the PDX model has contributed to increasing our knowledge of the molecular mechanisms and signaling pathways involved in the response of cancer cells to the treatments and the molecular mechanisms that are established in drug-resistance and molecular therapies [58]. In light of this, PCa PDX appears to be a good preclinical model for reproducing tumor histological features and maintaining the gene expression patterns and gene variants of patients [59].

A recent very comprehensive study, the MURAL (Murbourne Urological Research Alliance) collection, has reported a set of patient PCa-derived xenografts [60]. The collection included 59 PDXs established from 41 tumors of 30 patients, including treatment-naïve men and those with metastatic castration-resistant prostate cancer, representing the clinicopathological and genomic spectrum of PCa. The PDXs were characterized for inter- and intra-tumor heterogeneity by whole-bulk single-cell RNA sequencing, as well as by histology [60].

This cohort has allowed a fast and significant screening of numerous therapies, even in combination, with respect to a limited panel of certified and available PCa cell lines, by using a limited biological replicate number, which are then validated with targeted and less-dispersed standard preclinical experiments. The authors stratified these PDXs by phenotype and molecular characteristics by defining the subgroups. In fact, the aim of this project was not to establish “avatars” for the personalized therapy of selected patients but to create a robust in vivo biobank to identify promising, innovative, and even risky therapies to be tested in the preclinical phase. The work that the authors did to stratify the response to the various therapies tested with the background of gene alterations and RNA expression was very interesting [60].

Recently, several research groups have dedicated much effort to the formation of PDX biobanks to advance drug translationality from research to clinical application faster and with less bias. Very recently, a collection of five PDXs representing hormone-naïve, androgen-sensitive, CRPC primary tumors and neuroendocrine-differentiated tumors (CRPC-NE) has been established and histologically and molecularly characterized [35]. The most interesting results were the responses of the CRPC-NE model to PARP inhibitors and that these PDXs recapitulated disease progression from androgen-responsive CRPC, including CRPC with neuroendocrine (NE) features.

Despite the beauty of these projects, the great work of decades behind these results has also emerged. The data from the group of Béraud and colleagues represent a 10-year research project of the creation of the urological PDX, in which, of the 240 PCa originally implanted, the five present in the publication are the ones that survived [35].

Undoubtedly, this technology includes expensive methods that involve many researchers and technicians; to date, the PDX model is the one that has proved to be the most robust in significance and the fastest in translationality for the broad-spectrum screening of molecules and therapies.

By combining PDO and PDX with high-throughput sequencing technologies, researchers can perform comprehensive analyses of cancer biology, identify relevant biomarkers, and test potential therapeutic interventions in a personalized way. This approach has the potential to accelerate drug development, improve treatment outcomes, and move toward more personalized and targeted cancer therapies. However, it is essential to remember that research involving these models requires rigorous validation and integration with clinical data to ensure their clinical relevance and translational impact.

## 4. Current Application of PDO and PDX in PCa Precision Medicine

PDX and PDO models have the potential to revolutionize how we study and treat prostate cancer using precision medicine principles. Precision medicine is an approach to medical care and research that takes into account the individual variability of genes, the environment, and the lifestyle of each person, so the goal is to tailor medical decisions and treatments to each patient’s characteristics [37].

Patient-derived xenografts have emerged as an invaluable tool in the field of precision medicine, especially in the context of prostate cancer. PDX models better mimic the complexity and biological behavior of human tumors than traditional cell line models. This makes them a powerful platform that preserves the unique characteristics and heterogeneity of the original tumor and allows for the study of tumor growth, progression, and response to treatments in a context that closely resembles the patient’s cancer [34].

## 5. Personalized Drug Screening

The PDX and PDO models can be used to test a variety of treatments, including chemotherapy, targeted therapies, and immunotherapies. Organoids allow for high-throughput screening of various drugs to identify which ones are most effective against a patient’s cancer [61,62]. This may guide the selection of targeted therapies or combinations that have the potential for improved therapeutic outcomes.

An interesting androgen-dependent PCa patient-derived xenograft (PDX) model from treatment-naïve, soft tissue metastasis (PNPCa), and PDXOs have recently been developed to test therapy response [63].

The authors tested whether pre-existing therapy resistance in treatment-naïve PCa patients by developing PDX organoid derivation (PDXO) (Figure 1) to facilitate in vitro immunoassays and drug screening. Functional testing of targeted treatments according to the genomic profile of the PNPCa PDX and organoids was performed using PDX organoids from three different models [63]. In addition, organoids were tested for drug response to repurposing of FDA-approved compounds (74 compounds), including routinely used PCa standard-of-care drugs; as well, different FDA-approved drugs with indications for other cancer types were assayed. Interestingly, ponatinib, sunitinib, and sorafenib kinase inhibitors were highly effective on all PDX-derived organoids and patients from advanced cases with acquired resistance to standard-of-care compounds [63].

While sorafenib and sunitinib have been tested in Phase II/III clinical trials for CRPC (NCT00137436), ponatinib has not yet been validated for PCa [64,65]. Notably, from this drug-screening study, ponatinib emerged as highly effective in the metastatic PDX of PCa and PDOs, demonstrating significant tumor growth inhibition and a very good tolerability. These results strongly supported the potential of organoids-based assays on drug-screening applications in PCa.

In Table 2, we would like to summarize some examples of PDO and PDX models that we have discussed so far, established from PCa samples and cell lines, which over the last 10 years have provided relevant data about PCa progression mechanisms and about novel treatment approaches.

A procedure for the preparation of PDXO for prostate cancer (PCa) has recently been optimized for drug testing where the essential viability parameters of the ATP-based assay organoids are controlled [66]. The authors developed a confocal imaging pipeline to study the drug-induced effects on tumor growth and cell death at the single organoid level to provide drug response predictions as quickly as possible. Furthermore, these methodologies have attempted to provide standard operating procedures for organoid drug testing, which would allow for better comparisons between data from different treatments. In fact, testing different treatment options on patient-derived organoids can help predict which treatments are likely to be successful for an individual patient. This can minimize the trial-and-error process and improve the chances of selecting the most effective treatment upfront.

## 6. Drug Resistance

PDXs and PDOs can also be used to investigate mechanisms by which, after the initial benefit, PCa acquire drug resistance. Furthermore, a group of patients may not respond to therapy at all and develop novel resistance. By observing how the models respond to treatments over time, researchers can gain insights into why some treatments become less effective and identify strategies to overcome resistance [67]. At least three general mechanisms leading to the development of resistance in CRPC have been reported, such as mutations of the AR, ETS, TP53, and PTEN genes, chromosomal amplification and chromosomal rearrangement, and activation of bypass signals such as the receptor pathway of glucocorticoids that compensates for the loss of the AR signal [68]. Although the tissue integration rate of PCa PDX is still difficult despite the use of matrigel media (10–40% engraftment rate and extended latency times), some methods to successfully develop tissue-transplantable human PCa PDX have been shown to be effective by using the immunocompromised mice [69,70] to evaluate both anticancer efficacy and toxicity. In fact, it was initially demonstrated that 3D PDX PCa cells acquired an increased resistance to docetaxel compared to the cell lines commonly used in PCa research [69]. The further step forward in the translational research was achieved by the introduction of humanized mouse models, which allowed the recapitulation of an immunological system similar to the human one [71]. These mouse models represent an indispensable approach of PDX technology to validate new immunological treatments and preclinical personalized oncological therapies.

ERBB pathway dysfunction is universally recognized to be implicated in the progression of several cancers, including CRPC [72], and over the years, many molecules aimed at perturbing the stability of HER2/HER3 have been developed, such as pertuzumab and the latest generation of afatinib [73,74]. However, even these molecules after some time lose their functionality following the acquired resistance of the tumor to the drug. Recently, prostate cancer xenograft mouse models derived from CRPC patients with elevated erbB-3 receptor tyrosine-protein kinase (HER3) expression have been reported [75,76]. Gil and colleagues subcutaneously implanted metastatic lymph node biopsy subsections of a CRPC patient in intact non-obese diabetic (NOD) scid gamma (NSG) mice (NOD.Cg-Prkdcscid Il2rgtm1WjI/SzJ). Subsequent passages were undertaken in castrated mice. Organoids from PDX tumor tissue (PDXO) were then obtained [76]. The authors thus provided strong evidence that HER3 overexpression was associated with a shorter time to castration resistance development and worse overall survival. Importantly, in vivo targeting of HER3 signaling, strongly induced after castration, showed minimal impact on tumor growth, while in vitro targeting itself showed anti-oncogenic activity, probably because of the release of paracrine factors that the PDX model allows to act [76].

These results suggest that the experience of the PDX and PDO models manages to bypass the biases obtained from the in vitro clinical experimentation. Further studies on drug combinations incorporating a mix of HER3/ERBB inhibitors and androgen deprivation (ADT) and/or the co-targeting of inflammatory factors such as for IL-23 can be envisaged.

## 7. Biomarker Discovery

PDX and PDO models can aid in the identification of biomarkers associated with treatment response or resistance. By analyzing the genetic and molecular changes that occur in the models following treatment, it can identify potential biomarkers that can guide treatment decisions for patients with similar genetic profiles and especially for the patient matched with the model [76]. In this sense, Luo and colleagues recently reported the microarray data from two PDXs whose tissues are derived from single-patient PCa where LTL-313B and LTL-313H models exhibited low and high metastatic potential, respectively [77]. The transmembrane TMEM45B protein was identified as a candidate associated with PCa aggressiveness and metastasis. The authors highlighted how the tumor cell lines derived from the xenografts of the collected PCa patients preserved the characteristics of patients’ tumors by varying the in vivo metastatic capacity [77]. In addition to being models with important prognostic implications, they represent an important tool for studying the mechanisms of metastasis in PCa and for the discovery of new markers associated with the probability of developing metastatic processes. This study suggests the immediate translational prognostic value of TMEM45B in primary cancer as the detection of its expression can be immediately applied to patients undergoing radical prostatectomy or biopsy.

Other interesting studies involving the use of PDXs to identify novel and reliable biomarkers associable to the clinical outcomes of PCa patients led to the detection of AR splice variant-7 (AR-V7) as a predictive biomarker of AR-targeted therapy resistance in castration-resistant prostate cancer [78], the α2 chain of interleukin-13 receptor (IL13Rα2) highly expressed in castration-resistant prostate cancer PDXs and useful as serum biomarker [79], and the glucose transporter GLUT3/SLC2A3 as a biomarker of hypoxic prostate epithelial cells and prostate tumors [80], just to mention some of the most recent studies.

Castration-resistant prostate cancer is the most aggressive and lethal form of prostate cancer. Epithelial–mesenchymal plasticity (EMP) has been associated with CRPC progression. It has been reported that androgen deprivation therapy (ADT), despite good initial clinical responses, promotes EMP [81,82]. Some authors have tried to identify early markers of resistance to therapy by using liquid biopsy. Recently, rare CTCs have been isolated from the blood of PCa patients, showing an association between their number and progression free survival (PFS) and overall survival (OS) better than reported with PSA quantification [83,84,85].

Hybrid CTCs that simultaneously express epithelial and mesenchymal genes are a more aggressive cell population with a higher metastatic potential than fully epithelial or mesenchymal cells [86,87]. They also express markers of stemness and have self-renewing properties [87]. Hassan and colleagues recently developed a 42-gene RT-qPCR panel to detect CTCs with a hybrid EMP phenotype in PDX models of prostate cancer [88] showing significant correlations in gene expression between tumors and the respective CTC number. In a further study, Hassan and colleagues demonstrated elevated numbers of CTCs after the castration of mouse models with a more mesenchymal phenotype, even when the PDX tumors appeared fully regressed and with no detectable PSA, showing the presence of residual disease [89]. Some EMP-related genes were associated with castration resistance in tumors of PDX models, whereas the epithelial signature was only found in CTCs from uncastrated mice [89].

Unfortunately, the research for biomarkers in prostate cancer and PCa patient fluids by using organoid models is still lagging behind that with PDXs mentioned above and in PDO models obtained from other tumor types [90].

## 8. Application in Clinical Practice

In recent years, the development of high technology has favored the advent of targeted therapies, significantly modifying the clinical management of oncological treatment. However, their effectiveness often remains limited mainly because of the onset of acquired resistance to the treatment for which protocols have been developed that combine drugs targeting different signaling pathways [91,92,93]. In order to identify personalized therapy that is as reliable as possible, setting up an ex vivo model has become very important [44]. Since, in clinical practice, the time between diagnosis, possible surgery, and/or treatment decision is approximately two to three weeks, “avatar” models should be established as quickly as possible. With this time frame available, PDOs have emerged as an effective ex vivo molecule- and drug-screening tool for therapy decisions [44,94]. Numerous studies have highlighted the ability of PDOs to predict responses to clinical treatment in matched patients, even deciding to change therapy in time before the onset of drug resistance [93,94].

A very interesting study involving a platform called therapeutically guided multidrug optimization (TGMO) has developed a workflow that allows rapid identification of multiple personalized drug combinations [95]. The platform uses a minimum number of experimental tests on organoids of colorectal cancer (CRC) patients, exploiting an in silico mathematical prediction on the efficacy of the predicted drugs [96].

These innovative experimental platforms involving the use of PDOs to enable the optimization of multi-drug combination therapy tailored to individual patients are widespread in clinical trials of CRC, lung cancer, and ovarian cancer [97,98,99].

On clinicaltrials.gov, we searched for the keyword “organoid cancer” and identified 151 studies as of September 2023, many of which are currently recruiting. Sixty-two studies were interventional (clinical trials), and 89 were observational studies. The tumors taken into consideration were mainly digestive system tumors, breast cancer, blood tumors, lung cancer, ovarian tumors, and pancreatic cancers. Unfortunately, by adding the keyword “prostate” to the previous two, no type of clinical trials was registered. This finding supports the idea that the use of PDO models to refine precision medicine in PCa patients still lags behind the basic research previously discussed.

For prostate cancer, it is clear that there are many fewer personalized targeted therapies than for other tumors, despite it being an insidious and metastatic type of cancer, so there is a clear clinical need. The overriding challenge is that we do not understand the type of tumor microenvironment that PCa has and the factors that induce resistance to therapies. There are no models to effectively study this cancer and develop innovative therapies other than developing clinical studies in parallel with the genetic and molecular dissection of PDO.

## 9. General Challenges Addressed in the PDX and PDO Models and Conclusions

Prostate cancer, due to its histological nature and anatomical location, is a tumor with a high incidence of mortality, and compared to other types of tumors, it is still behind in the definition of biomarkers for the classification of molecular histotypes and the identification of specific therapeutic molecular targets.

The development of the PCa PDX and PDO models has provided an enormous amount of information on the biological basis that identifies the heterogeneity of prostate cancer over the last 15 years, becoming an invaluable resource for research and the optimization of therapy protocols targeted at prostate cancer and for the identification of increasingly specific biomarkers associated with the success of therapies. Prostate cancer lags behind other cancer types in marker-based classification for treatments.

PDXs and PDOs aim to replicate aspects of human cancer biology in a laboratory setting, but they have unique challenges and advantages. As discussed, both models help address the challenge of tumor heterogeneity [34,44,60,93]. Tumors often consist of a mix of cancer cell types with varying genetic mutations. These models can capture this diversity, allowing researchers to study how different cell populations respond to treatments.

In fact, one of the primary goals of PDX and PDO models is to predict how a patient’s tumor will respond to specific drugs. This is crucial for personalized medicine and for selecting the most effective treatment for individual patients. Overall, these models provide a platform for drug screening and testing to evaluate the effectiveness of various drugs and drug combinations on patient-derived cancer samples [69,70,73,74]. This can accelerate drug development by identifying promising candidates and avoiding ineffective treatments. Personalized medicine represents a significant challenge in cancer treatment. With this perspective, the PDX and PDO models contribute to the development of patient-specific treatment strategies, increasing the chances of positive outcomes.

Most importantly, the PDX and PDO models reduce the need for extensive animal testing in cancer research [56]. While they still involve the use of animals (xenografts), they minimize this by using patient-derived samples. In fact, the use of mouse models is still essential to maintain a high degree of biological relevance compared to traditional cell lines. Above all, PDXs better recapitulate the tumor microenvironment, allowing more accurate studies of cancer biology and drug responses.

We have also highlighted in this review that although PDX and PDO models have many advantages, they also have limitations, such as the cost and time required to establish and maintain them, potential genetic drift over time, and the inability to fully replicate the complex tumor microenvironment. Researchers continually work to overcome these challenges and improve the utility and accuracy of these models in cancer research and drug development.

In general, the PDX model is the one that still prevails in the most technological studies of cancer and that is most easily found in studies of PCa. This is due to the fact that many companies can provide certified mouse models of PDX with the clinical history of patients from whom the tumor has been removed. This has allowed many excellent basic research groups to investigate molecular aspects without directly having cohorts of patients, significantly increasing the scientific information in the field in a short time. Research based on PDOs, on the other hand, is a little further behind that of PDXs, and this is due to the fact that the human-cell-based technologies are still developing compared to those of animals, which have been well established for years in research institutes and accepted by all scientific journals.

For the future, it is hoped that there will be greater investments by government programs and companies for the funding of human-relevant methodologies to increase the development and applicability of PDOs and PDXs as standard methodologies in the study and treatment of tumors.

## Figures and Tables

**Figure 1 biomedicines-11-02743-f001:**
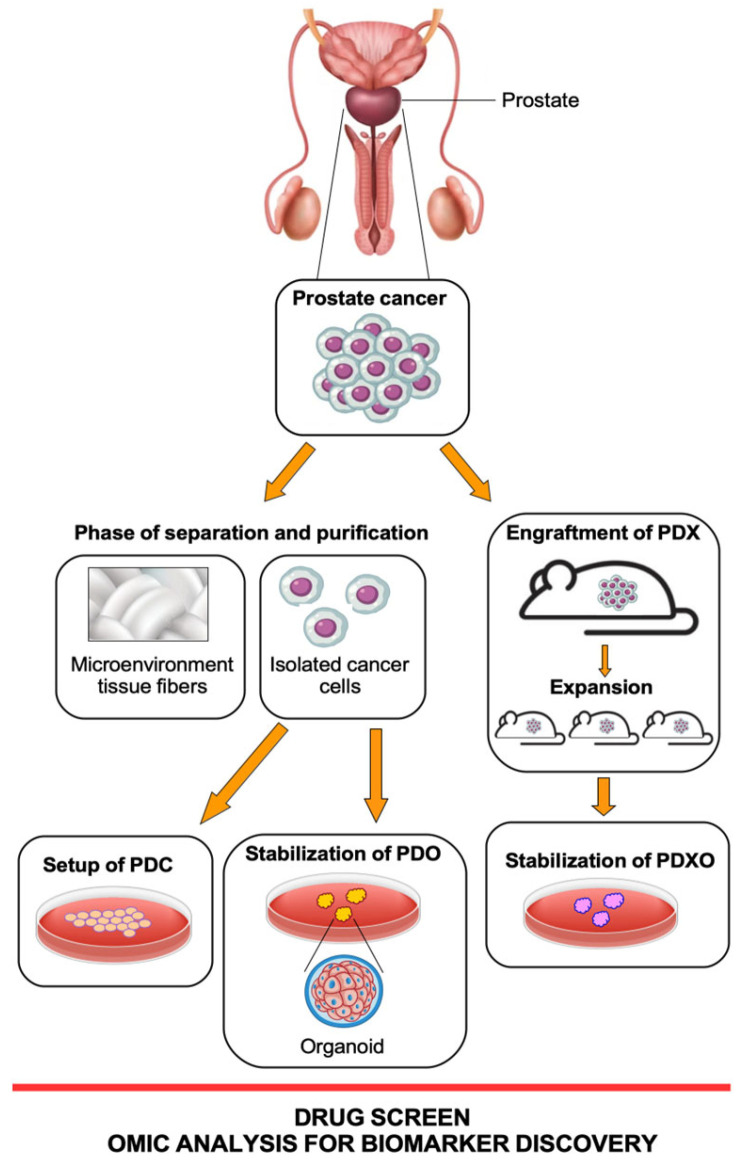
Schematic workflow of three patient-derived models: PDC (patient-derived cells), PDX (patient-derived xenografts), and PDO (patient-derived organoids). PDX-derived organoids (PDXO) are 3D in vitro models generated from patient tumor tissue that have been previously transferred into mouse models for expansion. All these models are derived from tumor tissue from the same PCa patient but throughout different approaches. Figure created at https://www.freepik.com/ (accessed on 1 August 2023).

**Table 2 biomedicines-11-02743-t002:** The examples of PDO and PDX models established from PCa samples and cell lines in the last 10 years discussed in this review.

Models	Tissues/Cells	Notes	References
PDO	Human biopsy samples and circulating tumor cells	The engraftment percentage was 15–20%. Characterization of PCa subtypes.	[37]
PDO	Human biopsy samples	The engraftment percentage was 16%. Characterization of neuroendocrine prostate cancer.	[41]
PDO	Healthy mouse and human prostate, human metastatic prostate cancer lesions, and circulating tumor cells	Development of the protocol for the engraftment of normal and tumoral tissues from the prostate. Characterization of the human and mouse organoids.	[38]
PDO	PCa specimens from a cohort of 81 patients with different pathological and clinical features	Morphological, immunohistochemical and genomic profiles of whole organoids to define the subtypes correlated to the PCa patients.	[40]
PDO	Localized PCa biopsies and radical prostatectomy specimens	Single-cell molecular analyses of established organoids to characterize the heterogeneity of tumor cells, subpopulations of epithelial cells, stromal cells, and tumor microenvironments.	[44]
PDX	PCa human cell lines: LNCaP, PC-3, Ca-2, DU145, VCaP	Set-up of the procedures for the generation of prostate cancer PDX models.	[20]
PDX	Setup of 80 PDXs derived from 47 human prostate cancer donors.	Some PDXs have generated cell lines to use as working models (MDA-PCa-2a and 2b). The histopathologic, genomic, and molecular characteristics are performed. Treatment with erdafitinib (FGFR inhibitor).	[34]
PDX	Setup of 5 PDX models from PCa	This collection included hormone-naïve, androgen-sensitive, and castration-resistant (CRPC) primary tumors, as well as prostate carcinoma with neuroendocrine differentiation (CRPC-NE). Morphological and immunohistochemical description and genomic profiles. Treatment with docetaxel, leuprolenin, enzalutamide, abiraterone, and olaparib.	[35]
PDX/PDXO	59 PDXs established from 41 specimens obtained from 30 PCa patients	Morphological and immunohistochemical description, genomic profiles, and gene expression profiles of MURAL cohort PDXs; 22 PDX tissues were grown as organoids to perform drug screening with apalutamide, enzalutamide, azacytidine, AZD1775 (Wee1 inhibitor), VX-970 (ATR inhibitor), docetaxel, carboplatin, and talazoparib. Drug as a single agent or combination.	[60]
PDX/PDXO	PDX model derived from a treatment-naïve soft tissue metastasis (PNPCa), with androgen-sensitive characteristics	Molecular characterization by DNA and RNA sequencing of PDX and establishment of PDXO to assess whether therapy resistance preexists in this treatment-naïve PCa case.	[63]
PDO/PDXO	Tissue from human PDX or prostate tissue from genetically engineered mouse model (GEMM) as source material to obtain individual cells	Assess the therapeutic potential of new drugs in the treatment of neuroendocrine prostate cancer (NEPC). Drug as a single agent or combination.	[36]

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
