# Peer review of "New Therapeutic Perspectives in Prostate Cancer: Patient-Derived Organoids and Patient-Derived Xenograft Models in Precision Medicine"

_biomedicines, 2023, doi:10.3390/biomedicines11102743_

Round 1

Reviewer 1 Report

The manuscript “New therapeutic perspectives in prostate cancer: patient derived organoids and patient-derived xenograft models in precision medicine” is an interesting review describing the rising importance of PDX and PDO models in the development of new anticancer therapies, especially against prostate cancer.

The article is well organized and comprehensively described, giving a wide introduction in the field and a clear presentation of data. Few corrections in the style of presentation and contents should be made to improve the quality of the manuscript. Hence, it is suitable for publication after the following minor revisions:

- In the abstract authors state to review “the most recent advances in the use the organoid culture system and PDXs in prostate cancer”, however they should clarify the range of time to which the studies described belong (for example the last decade) to help readers to get a perspective of what is included in the review.

- Provide a table/figure with an overview of all PDX and PDO models described in the text to summarize the data and give a more immediate data consultation.

- Minor english revision is necessary to correct several typos and misleading sentences. For example: line 148, “mice proved inadequate” should be changed in “mice proved to be inadequate”. Lines 149-150 and 317 should be rephrased. In line 525, change “n clinicaltrials.gov” with “in clinicaltrials.gov”

- When introducing a new acronym in the text, also write what it stands for. For example, in line 165, the extended meaning for the acronysm CNAs is missing.

- By contrast, what the acronyms (CRPC) or (PCa) stand for is repeated multiple times in the text, while they should be specified only the first time they appear in the text.

- Improve the outline of table 1, since in the Drawback words are stuck together.

- Line 395, after mentioning the phase II/III clinical trials for CRPC, please add the corresponding NCT from Clinical.gov to better identify the study.

Minor editing of English language required

Author Response

Reviewer 1

We thank the reviewer for the efforts made on the revision of our manuscript. We also thank for the valuable comments on our manuscript. In the revised version, we addressed reviewers' comments by responding point-by-point. We believe that the changes have significantly improved the manuscript and we hope to meet a positive opinion from the reviewer. We have marked all changes in the manuscript.

The manuscript “New therapeutic perspectives in prostate cancer: patient derived organoids and patient-derived xenograft models in precision medicine” is an interesting review describing the rising importance of PDX and PDO models in the development of new anticancer therapies, especially against prostate cancer.

The article is well organized and comprehensively described, giving a wide introduction in the field and a clear presentation of data. Few corrections in the style of presentation and contents should be made to improve the quality of the manuscript. Hence, it is suitable for publication after the following minor revisions:

Answer: We thank the reviewer for the general appreciation of the topic and the approach of the review. 

- In the abstract authors state to review “the most recent advances in the use the organoid culture system and PDXs in prostate cancer”, however they should clarify the range of time to which the studies described belong (for example the last decade) to help readers to get a perspective of what is included in the review.

Answer: We have clarified this point. We specified that we took into consideration the most significant works of recent ten years as every year the number of papers on this topic is increasingly higher, even if studies on organoids in prostate cancer proceed more slowly.

- Provide a table/figure with an overview of all PDX and PDO models described in the text to summarize the data and give a more immediate data consultation.

Answer: We thank the reviewer for the suggestion. We added new Table 2 reporting examples of PDO and PDX models established from PCa samples and cell lines, in the last ten years, discussed in the manuscript.

- Minor english revision is necessary to correct several typos and misleading sentences. For example: line 148, “mice proved inadequate” should be changed in “mice proved to be inadequate”. Lines 149-150 and 317 should be rephrased. In line 525, change “n clinicaltrials.gov” with “in clinicaltrials.gov”

Answer: We thank the reviewer for these specific observations. We have made the indicated corrections and revised the English throughout the text.

- When introducing a new acronym in the text, also write what it stands for. For example, in line 165, the extended meaning for the acronysm CNAs is missing.

- By contrast, what the acronyms (CRPC) or (PCa) stand for is repeated multiple times in the text, while they should be specified only the first time they appear in the text.

Answer: We have checked the entire text for the meaning of all the acronyms.

- Improve the outline of table 1, since in the Drawback words are stuck together.

Answer: We thank the reviewer for this thoughtful feedback. The format seems correct in our word file, however we have changed the size of the columns to improve the visibility of the "Drawbacks".

- Line 395, after mentioning the phase II/III clinical trials for CRPC, please add the corresponding NCT from Clinical.gov to better identify the study.

Answer: We added this information in the revised text, this study was registered as ClinicalTrials.gov NCT00137436.

Reviewer 2 Report

1.      Please write PCa in full in line 31, even though it has been defined in the abstract  as it has not been defined in the introduction.

2.      Table 1 – please add references that helped you form conclusions around each of the models

3.      Please prepare a table of PDX and PDO for PCa from the last 10 years with the following columns in their tables: Model used (mice/human), Cells/tissue used, duration of culture, Drugs screened, main findings.

4.      When you talk of biomarker discovery – it is worth mentioning the used of techniques like next generation sequencing in combination with organoids. There are several papers on the field but it is relatively new concept. You may read and cite the following to demonstrate your knowledge base in the field. ‘Organoid Models and Next-Generation Sequencing for Bone Marrow and Related Disorders. Organoids 2023.’

5.      Please add a paragraph on challenges faced in PDX and PDO to acknowledge the difficulties involved in the science behind this field.

6.      Please add a table of the evidence presented for section 8 – applications in clinical practice – from the last 10 years.

Minor editing of English language required

Author Response

Reviewer 2

We thank the reviewer for the efforts made on the revision of our manuscript. In the revised version, we addressed reviewers' comments by responding point-by-point. We believe that the changes have significantly improved the manuscript and we hope to meet a positive opinion from the reviewer. We have marked all changes in the manuscript.

  1. Please write PCa in full in line 31, even though it has been defined in the abstract as it has not been defined in the introduction.

Answer: We thank the reviewer for this observation. We have checked the acronyms in the revised text.

  1. Table 1 – please add references that helped you form conclusions around each of the models.

Answer: We have added the requested references in the Table 1.

  1. Please prepare a table of PDX and PDO for PCa from the last 10 years with the following columns in their tables: Model used (mice/human), Cells/tissue used, duration of culture, Drugs screened, main findings.

Answer: We thank the reviewer for the suggestion. We added new Table 2 reporting examples of PDO and PDX models established from PCa samples and cell lines, in the last ten years, discussed in the manuscript.

  1. When you talk of biomarker discovery – it is worth mentioning the used of techniques like next generation sequencing in combination with organoids. There are several papers on the field but it is relatively new concept. You may read and cite the following to demonstrate your knowledge base in the field. ‘Organoid Models and Next-Generation Sequencing for Bone Marrow and Related Disorders. Organoids 2023.’

Answer: We thank the reviewer for this observation. In the text we have described studies that used NGS analysis to characterize PDOs from the point of view of status gene and expression profiles (Ref. 44,60,86-89). Indeed, we agree with the reviewer that multi-omics analysis is necessary to categorize the histological subtypes of PCa into more specific cell clusters.

We have inserted the suggested reference in the text as novel [93].

  1. Please add a paragraph on challenges faced in PDX and PDO to acknowledge the difficulties involved in the science behind this field.

Answer: We accepted the reviewer's suggestion and rewrote the new paragraph 9 as "General challenges addressed in the PDX and PDO models and conclusions".

  1. Please add a table of the evidence presented for section 8 – applications in clinical practice – from the last 10 years.

Answer: In paragraph 8 "applications in clinical practice" we commented many types of cancer including lung, ovarian and colorectal cancer have a PDO and/or PDX model included in a clinical trial (94-99). We therefore referred to these trials regarding the type of application that could be developed for PCa as to date there are no clinical trials that have PDO or PDX models under study. We therefore think that a table cannot be shown.
